# GymHydro: An Innovative Modular Small-Scale Smart Agriculture System for Hydroponic Greenhouses

Cristian Bua [1], Davide Adami [2] and Stefano Giordano [1,2,*]

1   Department of Information Engineering, University of Pisa, 56122 Pisa, Italy; cristian.bua@phd.unipi.it
2   CNIT—Department of Information Engineering, University of Pisa, 56122 Pisa, Italy; davide.adami@cnit.it
*   Correspondence: stefano.giordano@unipi.it

**Abstract:** In response to the challenges posed by climate change, including extreme weather events, such as heavy rainfall and droughts, the agricultural sector is increasingly seeking solutions for the efficient use of resources, particularly water. Pivotal aspects of smart agriculture include the establishment of weather-independent systems and the implementation of precise monitoring and control of plant growth and environmental conditions. Hydroponic cultivation techniques have emerged as transformative solutions with the potential to reduce water consumption for cultivation and offer a sheltered environment for crops, protecting them from the unpredictable impacts of climate change. However, a significant challenge lies in the frequent need for human intervention to ensure the efficiency and effectiveness of these systems. This paper introduces a novel system with a modular architecture, offering the ability to incorporate new functionalities without necessitating a complete system redesign. The autonomous hydroponic greenhouse, designed and implemented in this study, maintains stable environmental parameters to create an ideal environment for cultivating tomato plants. Actuators, receiving commands from a cloud application situated at the network's edge, automatically regulate environmental conditions. Decision-making within this application is facilitated by a PID control algorithm, ensuring precision in control commands transmitted through the MQTT protocol and the NGSI-LD message format. The system transitioned from a single virtual machine in the public cloud to edge computing, specifically on a Raspberry Pi 3, to address latency concerns. In this study, we analyzed various delay aspects and network latency to better understand their significance in delays. This transition resulted in a significant reduction in communication latency and a reduction in total service delay, enhancing the system's real-time responsiveness. The utilization of LoRa communication technology connects IoT devices to a gateway, typically located at the main farm building, addressing the challenge of limited Internet connectivity in remote greenhouse locations. Monitoring data are made accessible to end-users through a smartphone app, offering real-time insights into the greenhouse environment. Furthermore, end-users have the capability to modify system parameters manually and remotely when necessary. This approach not only provides a robust solution to climate-induced challenges but also enhances the efficiency and intelligence of agricultural practices. The transition to digitization poses a significant challenge for farmers. Our proposed system not only represents a step forward toward sustainable and precise agriculture but also serves as a practical demonstrator, providing farmers with a key tool during this crucial digital transition. The demonstrator enables farmers to optimize crop growth and resource management, concretely showcasing the benefits of smart and precise agriculture.

**Keywords:** Internet of Things; smart agriculture; LoRa; NGSI-LD; hydroponic greenhouse; Node-RED; edge computing; smart lighting; PID controller

## 1. Introduction

In the realm of agriculture, the impact of climate change, marked by increasingly frequent extreme events, such as water bombs and prolonged droughts, has prompted a critical re-evaluation of natural resource utilization, particularly water. The need for

more resilient systems, ones that are impervious to weather fluctuations, has led to the development of innovative cultivation technologies. Notably, hydroponic cultivation techniques have emerged as transformative solutions, boasting the potential to curtail water consumption for cultivation by up to 95% [1]. Furthermore, the strategic use of indoor spaces, exemplified by hydroponic greenhouses, provides a sheltered environment for crops, shielding them from the unpredictable effects of climate change.

In hydroponic systems, plants thrive in a nutrient-rich water solution instead of traditional soil. However, the efficiency of hydroponic systems often hinges on human intervention, presenting a significant challenge. In response to this challenge, the Internet of Things (IoT) emerges as a pivotal platform, facilitating the integration of smart services that demand monitoring and remote control of hydroponic systems. This integration is crucial for managing diverse parameters that influence plant growth.

This paper introduces a pioneering hydroponic system conceived as a modular gym. This architectural innovation streamlines the development and deployment of new services without necessitating a complete system redesign each time additional smart devices or functionalities are introduced. We have designed a gym tailored for precision agriculture with a specific emphasis on hydroponic systems to illustrate this concept. This gym is optimized for an autonomous hydroponic greenhouse dedicated to year-round cultivation of tomato plants, with the flexibility for multiple cultivation cycles annually (e.g., in summer and/or winter), owing to the system's autonomy from weather conditions.

The proposed architecture for this smart hydroponic greenhouse empowers the monitoring of air temperature, air humidity, water levels under plant roots, and light intensity. This monitoring ensures the stabilization of environmental parameters, creating an optimal setting for the cultivation of tomato plants. All IoT devices deployed within the hydroponic greenhouse are seamlessly connected to edge computing, utilizing the new LoRa modulation at 2.4 GHz. This strategic choice is particularly pertinent, as greenhouses are commonly situated in rural areas, often plagued by poor or non-existent network connectivity. Sensor data undergo processing on an edge computing server and, specifically, light intensity is regulated through a proportional–integral–derivative (PID) controller. Additionally, Node-RED facilitates data processing in the edge cloud, providing tools to craft a graphical interface for farmers or users. This interface empowers them to remotely modify operational parameters using an intuitive and user-friendly app on their smartphones.

The structure of the paper is as follows: Section 2 conducts a comprehensive review of prior papers and projects related to this work. Sections 3 and 4 introduce the modular system architecture and the methodology employed for making design decisions, respectively. Section 5 delineates the system setup and presents experimental results. Finally, Section 6 offers concluding remarks on the presented work.

## 2. Related Works

In this section, we will provide an overview of relevant research papers and activities in the domain of smart agriculture systems, emphasizing their significance in the context of our work.

In [2], the authors employed two micro-controller unit (MCU) nodes equipped with various sensors, including soil moisture, air temperature, humidity, soil temperature, light, and a water pump. LoRa network interfaces operating at 868 MHz facilitate data transmission, with a LoRaWAN application employed for data monitoring. The study used a threshold comparator to control air and soil temperature and air and soil humidity. These types of controls surpassed the set threshold a total of 5 times during a day, with peaks of 34.59 °C when the threshold was 32 °C, similarly with humidity. A transmission range comparison was also carried out between three types of LoRa end devices with two configurations: spreading factor (SF) 7 and SF 10. The result showed that Heltec LoRa ESP32 with 1.5 dBi achieved the longest range, at 1000 m. In conclusion, the comparison of power consumption for sending 100 packets of 10 bytes at 50 m using LoRa versus

sending them via Wi-Fi was shown. The result suggested that Wi-Fi can consume as much as 2.5 times the power as compared to LoRa.

Another work [3] focused on a LoRa-based system for data collection and management in two distinct areas: a vegetable farm and a swallow bird nest farm. The authors deployed two MCU nodes transmitting messages to a gateway with LoRa modulation at 915 MHz and testing the LoRa transmission range in an urban area and in a rural area. The results showed that in the urban area, the maximum distance was 1800 m, while in the rural area, the testing was stopped after 3200 m due to geographical constraints; however, based on the results, the authors predicted that the LoRa system could still communicate over longer distances.

In [4], an IoT-based hydroponic system was proposed, enabling the monitoring and control of crucial operating parameters, such as pH, humidity, and temperature. Data from a single MCU were transmitted via a Wi-Fi module to a public cloud. A Blink app was integrated, allowing decision-making based on the monitored data. The experiment showed that when pH levels were low or high, plants could not absorb the nutrients properly; hence, monitoring and maintaining the pH level is essential for hydroponically grown plants. The pH value also informed about nutrient availability, through which we can identify deficiencies. When pH levels dropped below 5.5, it could cause copper and iron toxicity or magnesium and calcium deficiencies, and pH levels above 6.5 caused iron deficiencies. The authors also showed failure results when the pH level was out of the above range. It resulted in yellowing between leaf veins, brown leaf margins of the plant, and an increased period required for plant growth, which indicated a deficiency in plant growth. The conclusions after experimentation were that a proper combination of nutrients needs to be maintained, temperature and humidity need to be monitored daily, pH of the nutrient solution should be monitored and maintained, and growing plants using the NFT method on a large scale requires more investment.

The investigation of a traditional hydroponic farm, emphasizing plant growth without soil, was explored in [5]. The authors concentrated on describing some issues of traditional hydroponic cultivation. Errors are common in human activities, but they should not be overlooked. Automation emerged as the optimal solution to minimize errors and human resource utilization. Appropriate sensors allow for continuous monitoring of environmental parameters, water quality, and levels at regular intervals. Farmers can reduce labor and resource costs through automated monitoring of farm conditions. Automation guarantees increased output, cost reduction, and shorter lead times. Agriculture is directly influenced by changing climatic conditions, which can impact crop production rates. IoT technology allows us to receive real-time updates about climatic conditions, enabling us to make informed decisions and take appropriate actions in response. Data collected from sensors may not be suitable for storage in a conventional database due to storage limitations. An effective solution to this issue is utilizing cloud storage. Sensors gather data across various parameters, which are then analyzed and transformed into more comprehensible and informative insights using analytics tools. Their proposed system is composed of a Raspberry Pi 3 Board attached to a temperature sensor, pH sensor, water level sensor, and soil temperature sensor, which transmit data via Wi-Fi to the Ubidots cloud dashboard.

In [6], a greenhouse for tomato plants was detailed, utilizing Arduino UNO and sensors for pH, temperature, and humidity monitoring. Data were transmitted via an ESP8266 Wi-Fi module to the cloud with the ThingSpeak application, which uses MQTT and HTTP protocols to communicate with them. The authors also focused on tomato plant cultivation details: 90–120 days from the time of sowing, 45–55 days from the flowering period, the best temperature range of tomato plants was between 16 and 29 °C, the soil needs good draining capability, and the pH value should range from 5.5 to 6.8. In conclusion, the authors showed how unstable the growth parameters are during a day, compared to ideal growing conditions.

Another approach in [7] introduced a model for designing greenhouse farming using IoT. The model needs to have sensors, an IoT board, cloud, and a mobile app to visualize

monitored parameter trends. In their proposed system, they used temperature, humidity, pH, pressure sensors, and Raspberry Pi as an IoT board to communicate with the ThingSpeak platform in the cloud.

The experiment in [8] was conducted in six high-tech greenhouse compartments during a period of six months of cherry tomato growing. The primary goal of the greenhouse operation was to maximize net profit, by controlling the greenhouse climate and crop with AI techniques. The artificial lighting system consisted of six high-pressure-sodium (HPS) lamps and eight multi-spectrum controllable LED lamps. Power supply to the LED lamps was coupled with the power supply of the HPS lamps, meaning that LED lamps could only be used in addition to the HPS lamps. For control of natural light and energy saving, two types of inside moveable screens (LUXOUS 1547 D FR energy screen (Ludvig Svensson Inc., Kinna, Sweden) and OBSCURA 9950 FR W light-blocking screen (Ludvig Svensson Inc., Kinna, Sweden)) were present. Different sensors in the greenhouse collected data on climate and irrigation automatically and returned them via the process computer and the digital interface REST API. Staff in the greenhouse collected data on crop parameters manually and entered the observations on a tablet. Sensors were used for monitoring inside climate parameters and equipment status, such as: lamp status (on/off) of both lighting systems (HPS and LED), energy and black-out screen position, air temperature inside, heating pipe temperature, heating power used for both heating systems, air absolute humidity inside, and $CO_2$ dosage. Net profit was calculated based on income minus costs. The income was determined from the kg of tomato fruits harvested x price per kg of fruits and fruit quality. The price depended on fruit quality and on the season. Resource use of electricity, heating, $CO_2$, water, nutrients, and labor were measured during the experiment per greenhouse compartment and multiplied by the given price: electricity on-peak price (07:00–23:00 h) = EUR 0.08 per kWh and off-peak price (23:00–7:00 h) = EUR 0.04 per kWh, heating price = EUR 0.03 per kWh, $CO_2$ price = EUR 0.08 per kg up to 12 kg/m$^2$ and EUR 0.20 per kg above 12 kg/m$^2$, and labor for crop maintenance = EUR 0.0085 per stem per m$^2$ per day. The results showed that the mean total cost was EUR 33/m$^2$ and the net profit ranged from EUR 3.10 to 6.86/m$^2$. The influence of light availability was investigated by increasing or decreasing the number of lighting hours by a maximum of 3 h per day, not changing the applied light intensity. An increase in lighting will result in higher electricity costs, higher production, and a little less cost for heating. A decrease of the number of lighting hours will generally have the opposite effect. It can be observed that the response of net profit on the application of light was much stronger than the response of changes in $CO_2$ dosing and temperature. The strong effect of light on net profit in the performance analysis requires a somewhat deeper analysis. Because of the large increase in the amount of natural sunlight and the reduction of product prices toward summer, the revenue of additional artificial light can be expected to differ during the growing season (December to May). To analyze the effect of artificial light in time, the effect of two additional hours of artificial lighting per day was added, for each week during the growing season, while maintaining the illumination for the other weeks unchanged. The additional 2 h of artificial lighting over 1 week resulted in an increase in net profit of around EUR 0.1/m$^2$ when applied around the tenth week after planting and in a strong drop toward the end of the growing period.

In [9], a Blockchain-enabled optimal smart contract for an autonomous greenhouse environment was presented. The proposed smart contract is based on a prediction, optimization, and control module for effective management and planning of an energy-efficient greenhouse environment. The greenhouse emulator was designed and implemented for collecting greenhouse sensors' data, whereas the greenhouse interface processes the collected data for controlling the actuator and preserves the data in the blockchain network. The proposed optimization approach was evaluated and compared with the baseline and optimization approaches. The analysis showed that the proposed approach led to a desired greenhouse environment, leading to an improved crop production and reduction in energy consumption (41% reduction in energy consumption, as compared to the baseline

approach, and 19% improvement compared to the optimization approach). Furthermore, the blockchain integration led to the following advancements in the traditional greenhouse, such as availability, scalability, throughput, privacy, and off-chain data storage. Finally, the Hyperledger caliper was used to assess the performance of the optimal smart contract in terms of latency, throughput, and resource utilization.

In [10], data collection was studied to reduce data redundancy for a critical event and ensure the latency constraint and main information in smart agriculture with consideration of the edge computing and SDN. First, from the perspective of event-driven sensing, a four-layer framework for smart agricultural IoT was introduced. Then, a three-step strategy was proposed for effective data collection in smart agriculture. In the first step, the MI from a historical dataset was used to sort the related sensing data types of different events. In the second step, the event identification based on edge computing was conducted by computing the minimum variance of the sensing data. Moreover, a data sensing method for collecting the most related data type was used to meet time constraints. Finally, the feasibility of the proposed strategy was verified by the experiments in a greenhouse. The results demonstrated that the proposed strategy provided a larger margin in balancing between data validity, energy consumption, and latency. The results demonstrated that the proposed strategy provided a larger margin in balancing between data validity, energy consumption, and latency.

The study in [11] called for more attention in academia to provide corresponding theoretical and methodological support for emerging financial, operation, and management issues. Technological advances providing cheaper sensors and safer networks will boost further application, but the drive must be hastened by overcoming the challenges faced to match the pace of population growth. One obvious challenge is how to make farmers as interested in implementing IoT systems as they are in high-yielding seeds and high-efficiency machines. Lifecycle IoT-based agriculture is necessary to solve this challenge by helping farmers recognize the quality of agriculture ingredients, improve the yields, as well as the quality, and produce creditable agriproducts for the market. Besides technical issues, emerging financial, operation, and management issues are gradually observed in the digitization of agriculture using IoT techniques. Innovative farm production modes and new types of agribusiness enterprises will arise to solve these issues.

This paper aims to introduce a different architecture for hydroponic greenhouse systems to reduce human intervention and to make the system easy to expand to include new features born over the years. Multiple MCU nodes are employed for monitoring environmental and water parameters, including temperature, humidity, water levels, and the intensity of visible-light wavelengths. The latter serves as input for a PID controller running on an edge computing server, with the algorithm's output generating PWM signals for four high-power LED lights that illuminate tomato plants, ensuring stable light intensity at visible wavelengths to optimize plant growth, even if sunlight or external light sources are low or absent. Furthermore, the system architecture stands out for its flexibility, enabling the incorporation of new devices and functionalities without necessitating a complete redesign to provide a continuous increase in devices or services in the greenhouse over the years. Finally, we advocate for the use of IoT with LoRa interfaces at 2.4 GHz [12].

## 3. System Architecture

The potential of IoT technology is imperative [13] for transitioning a conventional hydroponic greenhouse into an advanced, smart, and intelligent system. This section delineates the system architecture refer to Figure 1, emphasizing a design that facilitates the rapid creation of new services, while ensuring robustness, low power consumption, long-range communications, and reliability.

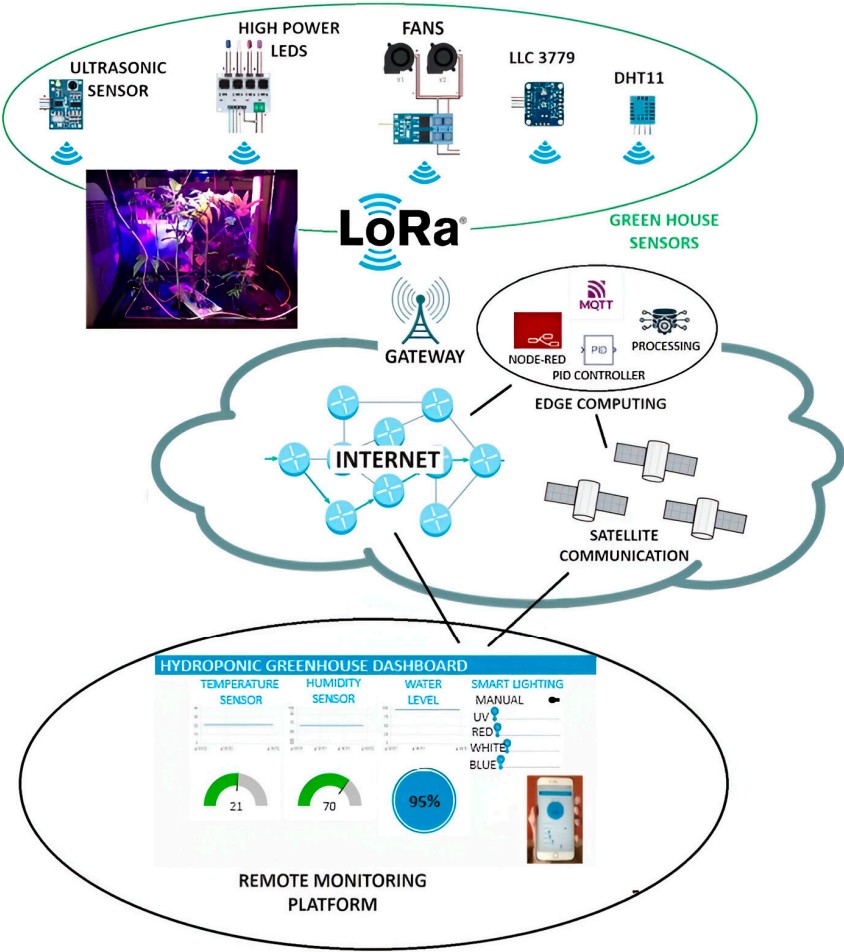

**Figure 1.** High-level system architecture.

The system architecture adopts the Node-RED low-code/no-code paradigm to streamline the integration of new hydroponic greenhouse sensing and actuating devices. This approach not only expedites the development of innovative monitoring and control functions but also enhances the system's adaptability to evolving requirements.

Devices within the hydroponic greenhouse connect to a LoRa gateway operating at the latest 2.4 GHz frequency band. This gateway, strategically located at the main farm building where Internet connectivity is typically available, ensures seamless communication and data exchange.

At the application level, the Message Queuing Telemetry Transport (MQTT) serves as the messaging protocol between hydroponic greenhouse devices and edge computing. Adopting a publish/subscribe paradigm and facilitated by a broker, MQTT enables efficient data exchange between the hydroponic greenhouse devices and the Node-RED server in edge computing.

For the data format, the New Generation Service Interfaces for Linked Data (NGSI-LD), standardized by ETSI, are employed. This format, in conjunction with the MQTT protocol, enables incoming data to be received by a broker, forwarded to the Node-RED server, and processed for actuating decisions, if required. The utilization of NGSI-LD adds a crucial layer of interoperability to the system, ensuring seamless communication and interaction between various components and other systems, thereby enhancing the overall compatibility of the proposed architecture.

The Node-RED server, featuring a cutting-edge PID controller, meticulously processes incoming data and executes actuating decisions. This groundbreaking addition ensures pinpoint control over hydroponic greenhouse devices, particularly high-power LEDs. Sub-

sequently, these decisions are seamlessly transmitted back to the broker and facilitated by the LoRa gateway, effectively communicated to the specific devices within the greenhouse. This innovative use of a PID controller for the precise regulation of lights stands as a key advancement in the proposed system architecture.

The system incorporates edge computing to enable real-time data processing at the network's edge. This ensures swift decision-making and reduces latency in controlling hydroponic greenhouse parameters. The combination of edge computing and the Node-RED server enhances the overall system efficiency.

End-users are granted access to a user-friendly Node-RED dashboard, enabling real-time monitoring of data and seamless control over the hydroponic greenhouse state. This intuitive interface empowers users to efficiently optimize their workflow, allowing them to visualize crucial parameters and make decisions remotely using their smartphones.

## 4. Methodology

In this section, we describe in more detail the components and the methodology adopted to implement the system.

As already highlighted, the IoT devices utilize the LoRa modulation at 2.4 GHz by exploiting the capabilities of the newest SX1280 Semtech chip (Semtech, Camarillo, CA, USA). LoRa offers better performance in terms of power consumption [14] and robustness to interference, such as coexistence with Wi-Fi. The authors of [15] demonstrated that the immunity of LoRa to Wi-Fi under co-channel coexistence scenarios is high, especially with a combination of low and high SF values. In-band coexistence scenarios for LoRa, especially in the case of the IEEE 802.11n interfering signal, result in an increasing C/I value for LoRa. A 250 kHz guard band between Wi-Fi and LoRa RF spectra is enough to ensure error-less LoRa communication. LoRa also offers better performance with respect to other technologies, such as ZigBee (the authors of [16] tested ZigBee range communication in outdoor line-of-sight (LOS) and non-line-of-sight (NLOS) conditions, achieving ranges of up to 100 m and 40 m, respectively), SigFox, and IEEE 802.11ah (Wi-Fi Halow). Additionally, LoRa provides longer range communications (up to 10 km in line-of-sight) [17] than other low-power communication technologies (up to 1 km) [18]. Greenhouses are typically situated in rural areas with poor or absent network coverage; however, thanks to LoRa, there is the possibility to transmit data to a gateway and, after that, to exploit other existing networks, such as 5G, LTE, Ethernet, or Optical Fiber. LoRa modulation works at more frequencies in the Industrial, Scientific, and Medical (ISM) band: 433 MHz, 868 MHz in Europe, and 915 MHz in the US, and recently, at 2.4 GHz with the newest SX128x chip category (Semtech, Camarillo, CA, USA). We propose the use of the frequency band at 2.4 GHz to use a unique chip across the world and to make the system usable anywhere, without special configurations. LoRa sub-GHz devices are limited in terms of the number of transmissions, since 433 MHz, 868 MHz, and 915 MHz bands are regulated by ETSI with a duty cycle between 1% and 10%, as well as in terms of globalization of the system, because every country has different frequency bands [19]. Instead, with a 2.4 GHz band, it is possible to transmit continuously without requesting specific authorizations and without paying money to take authorizations.

To demonstrate the benefits of a modular approach, at the beginning, we used a single MCU node connected to a LoRa gateway to obtain information from the sensors and send commands from the Node-RED server to the actuators. The MCU node had the STM32 chip (STM, Geneva, Switzerland) and the SX1280 module to communicate with the gateway. The MCU node was connected to four high-power LEDs (UV, Red/Blue, Blue, and White), one temperature and humidity sensor (DHT11 [20]), one ultrasonic sensor (HC-SR04 [21]), one spectrometer (Adafruit Industries LLC 3779 [22]), and two fans. Each high-power LED had 10 W of power and its light intensity was controlled by a PWM signal. The ultrasonic sensor was used to control the water level under plants. Two fans were used to emulate the wind: this was necessary to increase the robustness of the plants' stems and is also useful during pollination. A spectrometer monitored the light intensity at any visible wavelength.

This information was given as input to a PID controller, which allows for controlling the lighting obtained from the four high-power LEDs.

Subsequently, the system transitioned from a single MCU node to multiple nodes, significantly enhancing its capability to address real-world use cases. This evolution exemplified the system's inherent flexibility, demonstrating its ease in accommodating new services and devices for monitoring additional parameters or incorporating extended features. Each type of sensor became associated with a distinct MCU node, and all MCU nodes were equipped with the SX1280 Semtech chip for communication with the LoRa gateway. This architectural decision was driven by the necessity in real-world scenarios, where sensors are often positioned at various locations, and they may be situated at a considerable distance from the gateway. Figure 2 illustrates the block diagrams of the MCU nodes.

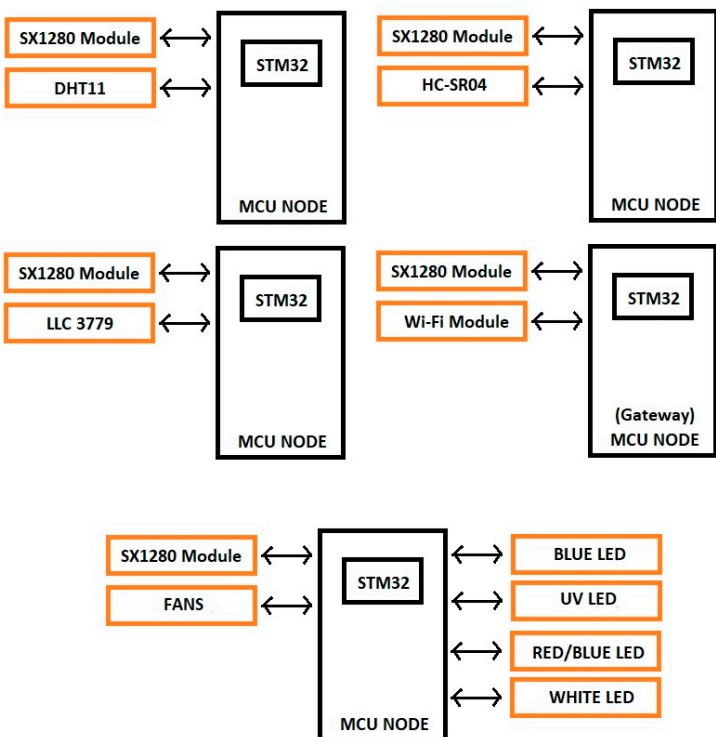

**Figure 2.** MCU nodes block diagram.

As highlighted in Section 3, the IoT devices in our hydroponic greenhouse system communicated seamlessly with the Node-RED server at a high level, utilizing the MQTT protocol and adopting a publish–subscribe paradigm. Initially, the broker was deployed in the public cloud but, aiming for improved performance and a quicker response to commands, we strategically migrated the computing to the edge, implementing it as a virtual function that embodies Mosquitto. In the publish–subscribe paradigm, clients operate without knowledge of each other, with each client capable of acting as a publisher, subscriber, or both. This unique feature of our system architecture facilitates the effortless addition of new IoT devices without necessitating any changes to the configuration of existing devices within the greenhouse. Introducing a new smart device defines a distinct message context, referred to as a "topic" (e.g., hydro-greenhouse/Pisa/tomato room/lamp), which the broker maps to a specific device. In this setup, the Node-RED server and the MCU node with LEDs function as both publishers and subscribers. On the other hand, the remaining MCU nodes exclusively operate as publishers. The gateway collects environmental data from sensors and directs them to the broker while awaiting commands for the actuators. The Node-RED server receives NGSI-LD format messages from the broker, processes them, and determines the commands to be published back to the broker

for the specific IoT devices. For instance, when a new device is added, it sends messages with a different topic, requiring only the addition of a new flow to the Node-RED server to process its data. If there is a need to add a new function to command an existing IoT device, creating a new flow on the Node-RED server and modifying the firmware of the specific IoT device to process new commands become necessary. This modular and flexible approach ensures the scalability and adaptability of our system.

NGSI-LD is an information model to support linked data [23]. The main constructs of NGSI-LD are entity, property, and relationship. Properties are a combination of an attribute and its value. Relationships allow to establish associations between entities using linked data. Typically, the NGSI-LD message formats have only one or two levels of property or relationship. JSON-LD, a JSON-based serialization format for linked data, allowed to develop this new format, expanding JSON in terms of URIs to define terms unambiguously. Figure 3 shows an example of a typical message format used in our system when a temperature sensor generates a message.

```json
{
        "id": "urn:ngsi-ld:AirTemperatureObserved:GreenHouse:0bsv4567",
        "type": "AirTemperatureOdserved",
        "dateObserved": {
                "type": "Property",
                "value": {
                        "@type" : "DateTime",
                        "@value" : "2023-05-05T12:00:00Z"
                }
        },

        "Temperature": {
                "type" : "Property",
                "value" : 22,
                "unitCode" : "Celsius"
        },

        "refPointOfInterest": {
                "type": "Relationship",
                "object": "urn:ngsi-ld:PointOfInterest:GreenHouse:Main"
        },

        "@context": [
                "https://schema.lab.fiware.org/ld/context",
                "https://uri.etsi.org/ngsi-ld-core-context.jsonld"
        ]
}
```

**Figure 3.** NGSI-LD message format.

OneM2M [24] is another message format that could be used. OneM2M is a global initiative that brings together the major ICT companies and aims to define an interoperability standard, including interfaces, APIs, and security guidelines. It features a common service layer, represented by a middleware that mediates the communication between use-case-specific hardware components and the oneM2M ecosystem. OneM2M provides a common service layer that includes a description of the service aspects.

However, what brought us to choose NGSI-LD is that oneM2M does not have such an exhaustive description for interacting with oneM2M devices/resources.

### 4.1. Smart Lighting with PID Controller

In [25], the authors conducted comprehensive experiments to identify the most optimal light wavelengths for the germination and growth of plants, with a particular focus on tomato plants. Their findings indicated that, for tomato plants, the most beneficial

wavelengths were found in the ranges of 660–730 nm, 420–500 nm, and the full spectrum. This insight into the specific light requirements for tomato plants served as a foundational reference for our hydroponic greenhouse system in Figure 4 [26].

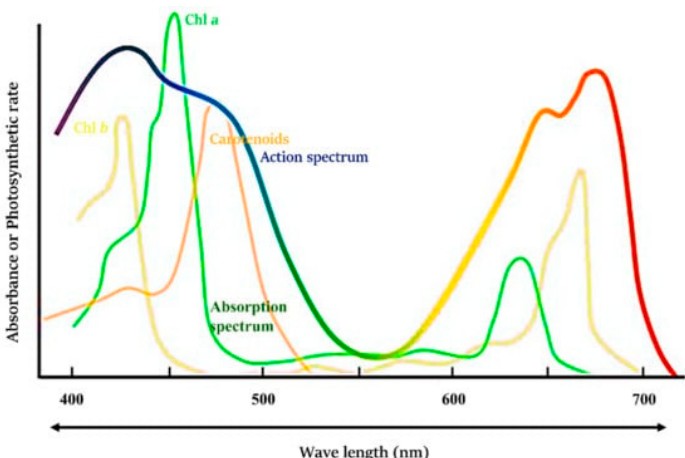

**Figure 4.** Plant absorption wavelengths.

In our system, we strategically employed four high-power LEDs to meet the unique light requirements of plants and sampled the visible spectrum every minute, assuming that sunlight varies significantly depending on cloud movements. The lighting configuration comprises a white LED that emits a full spectrum of light, a UV LED with a central wavelength ranging between 395 nm and 410 nm, a red/blue LED with central wavelengths at 455 nm and 640 nm, and a blue LED with a central emitting wavelength at 462 nm. These wavelength selections align with the scientifically proven effectiveness of different light ranges for supporting plant processes. Wavelengths falling between 420 nm and 315 nm are crucial for facilitating chlorophyll absorption. Meanwhile, the range between 420 nm and 500 nm, as well as between 620 nm and 750 nm, correspond to the blue and red LEDs, respectively. These specific wavelengths play a fundamental role in the absorption of chlorophyll and carotenoids, significantly impacting the efficiency of photosynthesis, a critical process for plant growth and development in our hydroponic greenhouse system. The four high-power-intensity LEDs collectively provide a lux level of 32,000, considering the lumens of the LEDs and prototype dimensions. In our scenario, we have implemented smart lighting to consistently maintain this 32,000 lux, even when external lighting conditions fluctuate. We connected the LEDs and the spectrometer to the Node-RED server to implement smart lighting, as previously described. In the Node-RED server, both the high-power LEDs and the spectrometer are abstracted as a single virtual IoT device. Another node in Node-RED corresponds to the PID controller, responsible for regulating the intensity values of the individual LEDs through PWM signals. Control commands are then transmitted to the MCU nodes, each identified by a unique MQTT topic.

Figure 5 shows the Node-RED flow diagram for controlling the actuators of the smart lighting system.

In manual mode, users have the flexibility to adjust light intensity using sliders, either based on personal experience or information gleaned from the charts on the dashboard. The light intensity values are received by a block of nodes and subsequently forwarded to the relevant actuators within the smart lighting system. This feature provides users with direct control over the greenhouse's light intensity. Upon switching to automatic mode, data from the spectrometer undergo separate processing by two PID nodes: one for the wavelength at 450 nm and the other at 650 nm. Each PID node calculates precise control signals based on the received data and the desired setpoints for the respective wavelength. A PID controller comprises proportional (P), integral (I), and differential (D) units, making it particularly applicable to basic linear and dynamic systems.

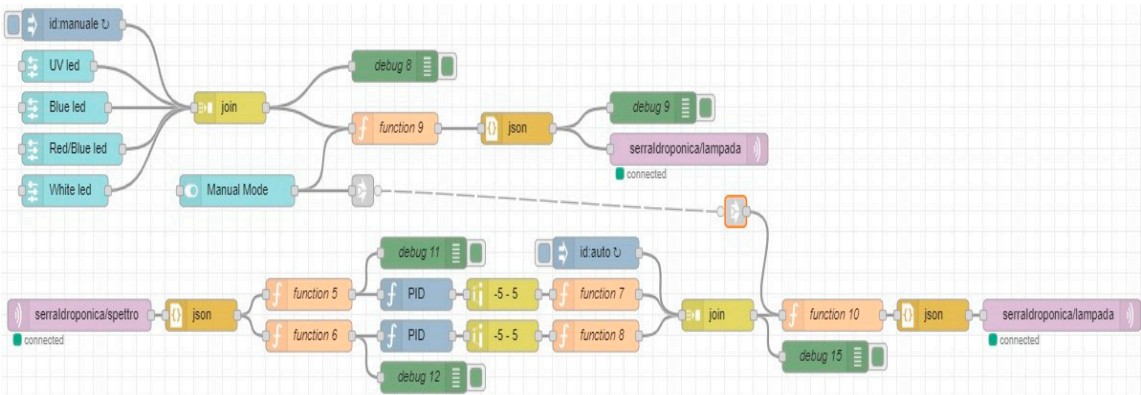

**Figure 5.** Data flow: smart lighting control.

A PID controller is a common feedback loop component in industrial control applications. This controller uses the data collected from the spectrometer to compare them with a reference value, which is used to calculate the new input value. The PID controller is very different from other control algorithms, such as the threshold comparator or the Kalman filter, because it makes the behavior of the system more accurate and stable by adjusting the input value according to the historical data and differences.

$$u(t) = K_P * e(t) + K_I * \int_0^t e(\tau)d\tau + K_D * \frac{de(t)}{dt} \tag{1}$$

Equation (1) describes the algorithm implemented by the PID controller node on the Node-Red server, whereas Figure 6 shows the block diagram with three main blocks: proportional, integral, and differential.

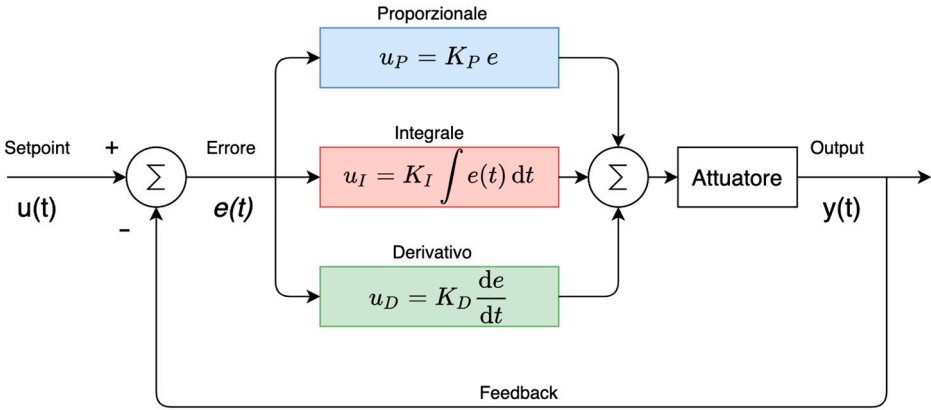

**Figure 6.** PID block diagram.

For the closed control loop of smart lighting at 450 nm (blue LED) and 650 nm (red LED) wavelengths, we utilized the PID algorithm function on the Node-RED server, implemented by the Node−RED−contrib−pid 1.1 tool [27]. The algorithm implemented in Node-Red, to be configured, requires four parameters: proportional band (PB), integral time (TI), derivative time (TD), and setpoint. To understand how these four contributions affect the algorithm, Equation (1) is rewritten as Equations (2) and (3):

$$u(t) = K_P * \left( e(t) + \frac{K_I}{K_P} * \int_0^t e(\tau)d\tau + \frac{K_D}{K_P} * \frac{de(t)}{dt} \right) \tag{2}$$

$$u(t) = \frac{1}{BP} * \left( e(t) + \frac{1}{T_I} * \int_0^t e(\tau)d\tau + T_D * \frac{de(t)}{dt} \right) \tag{3}$$

In our context, $e(t)$ signifies the disparity between the setpoint, and the value obtained at the specific wavelength by the spectrometer. The PID controller implemented does not operate in continuous time but rather in discrete time, because the intensity of wavelengths measured by the spectrometer is sampled and transmitted every Ts seconds, rather than continuously.

Several considerations and modifications were made to Equation (1). The signal processed is a discrete signal, $x[kTs]$, where $Ts$ is the sampling time, in our case, corresponding to the periodicity of light intensity measurements. If $e(t)$ is the error signal and $u(t)$ is the output of a PID controller, we can express Equation (1) as follows:

$$u(t) = K_P * e(t) + K_I * \int_0^t e(\tau)d\tau + K_D * \frac{de(t)}{dt} = u_P(t) + u_I(t) + u_D(t) \qquad (4)$$

Thus, for the discrete proportional component, we can obtain:

$$u_P[kTs] = K_P * e[kTs] \qquad (5)$$

The integral component can be evaluated based on Figure 7.

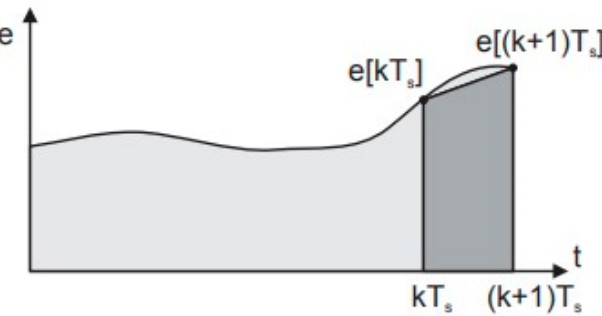

**Figure 7.** Discreate integration.

We can write:

$$u_I[(k+1)Ts] \cong u_I[kTs] + K_I\frac{Ts}{2}(e[(k+1)Ts] + e[kTs]) \qquad (6)$$

The derivative component can be obtained by recognizing that integration is the inverse operation of differentiation. Therefore, for the differentiation Equation (6), it can be rewritten as:

$$e[(k+1)Ts] \cong e[kTs] + K_I\frac{Ts}{2}(u_I[(k+1)Ts] + u_I[kTs]) \qquad (7)$$

$$u_D[(k+1)Ts] \cong K_D\frac{2}{Ts}(e[(k+1)Ts] - e[kTs]) - u_D[kTs] \qquad (8)$$

In the end, we obtain the discrete function of the PID controller, expressed as:

$$u[(k+1)Ts] = u_P[(k+1)Ts] + u_I[(k+1)Ts] + u_D[(k+1)Ts] \qquad (9)$$

*4.2. Dashboard*

We employed the Node-RED Flow-Based Programming (FBP) tool, developed by IBM's Emerging Technology Services team and currently under the OpenJS Foundation, to create an IoT application with web services. The process of developing software with Node-RED v12.22.9 is remarkably straightforward, eliminating the need for coding and allowing intuitive programming with minimal operations. Node-RED offers a browser-based GUI, utilizes lightweight protocols, encapsulates functions as nodes, and is easily accessible via the cloud. In this project, Node-RED was utilized to establish a virtual IoT device that processes data from various sensors within the hydroponic greenhouse. Smart lighting, for

example, consisting of four high-power LEDs and one spectrometer, is presented as five physical IoT devices. Similarly, temperature and humidity web services are depicted as two separate virtual IoT sensors on the dashboard, although they correspond to a single physical sensor (DHT11) in the system. Additionally, Node-RED showcases virtual IoT devices and sensors in the hydroponic greenhouse as web services through the dashboard.

The dashboard incorporates graphs, facilitated by the Node-RED-dashboard 3.4.0 plugin based on Angular v1. It includes on–off switches and sliders to enable the visualization of monitored data and interaction with the hydroponic greenhouse, specifically with smart lighting devices. Figure 8 shows the GUI of our GymHydro. Our aim was to demonstrate how the approach used in our work makes it easy to implement simple and intuitive dashboards, with the ability to modify them easily based on individual user needs.

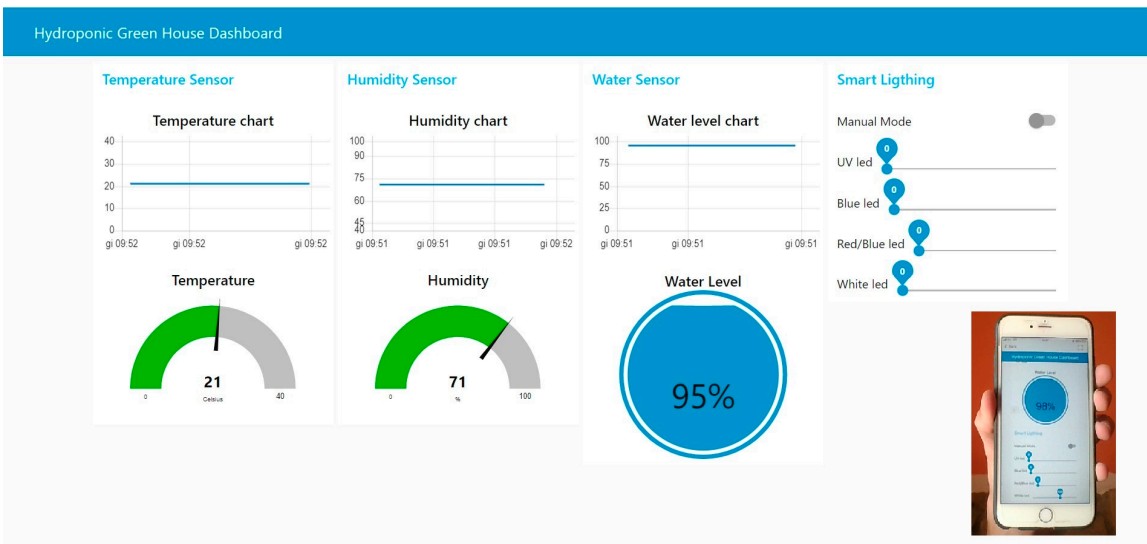

**Figure 8.** System dashboard.

## 5. System Setup and Functional Validation

The hydroponic greenhouse, measuring 50 cm × 40 cm × 50 cm, features a water container (32 cm × 22 cm × 7 cm) equipped with four liters of water and nutrient solutions. Within the container, a water pump facilitates the circulation of water beneath the plant roots. Positioned adjacent to the tomato plants are the DHT11 sensor and the LLC 3779 spectrometer. Above the plants, four high-power LEDs and two fans contribute to the environmental conditions.

Data collection was conducted using the spectrometer at wavelengths of 450 nm and 650 nm for the establishment of PID algorithm setpoints. These data were gathered with all four LEDs operating at maximum intensity and in the absence of external light sources. The resulting information was instrumental in fine-tuning the PID algorithm for optimal performance.

In Figure 9, the circuit diagram of the GymHydro sensing and actuating system is presented, providing a visual representation of the utilized hardware components.

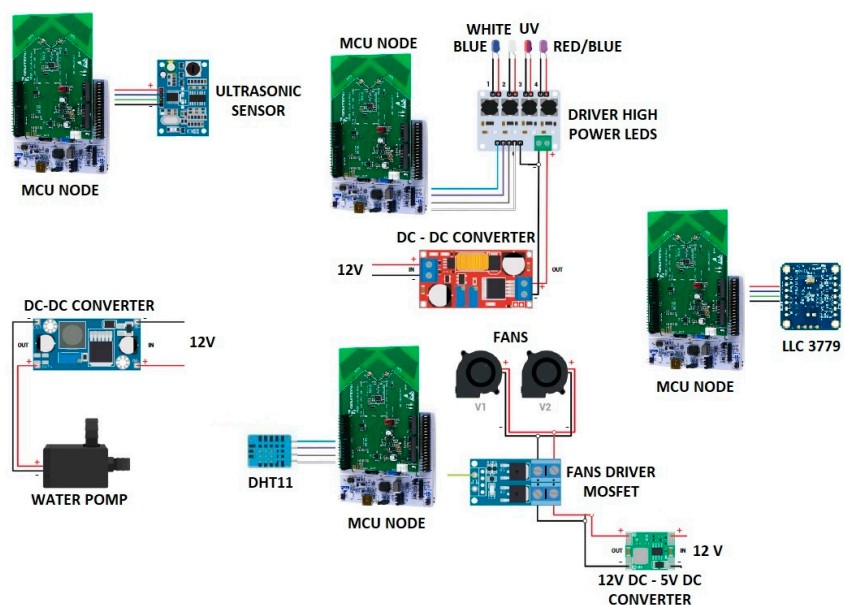

**Figure 9.** Circuit diagram of the proposed system.

### 5.1. Process Model Extraction for PID Controller Optimization

Our initial step involved extracting the process model to optimize the PID controller parameters, such as proportional band, integral time, and derivative time. In this context, the process is defined as the variation of light intensity in the system concerning the PWM signal provided to the LEDs. We assume that the model is linear:

$$f(x) = \begin{cases} mx + q, 0 < x < T \\ 0, otherwise \end{cases} \tag{10}$$

In the equation, '$f(x)$', where '$x$' represents the PWM signal provided, '$m$' is the angular coefficient, '$q$' is the y-intercept, and '$T$' is the maximum PWM signal possible (in our case, 255). The linearity of the process was affirmed through a linear regression analysis of light intensity measurements at 450 nm and 650 nm, as depicted in Figure 10.

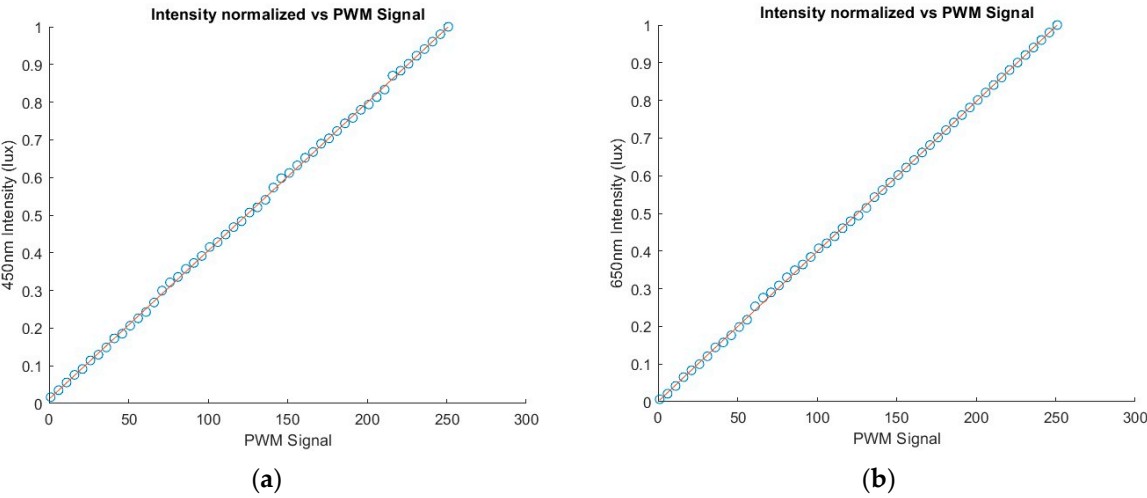

(**a**)　　　　　　　　　　　　　　　　(**b**)

**Figure 10.** (**a**) A graph illustrating measurements of wavelength intensity normalized at 450 nm with linear regression, and (**b**) a graph presenting measurements of wavelength intensity normalized at 650 nm with linear regression.

Table 1 provides the angular coefficients and y-intercepts of the intensity measurements.

**Table 1.** Linear regression coefficients.

| Wavelength (nm) | m | q |
|---|---|---|
| 450 | 0.004 | 0.02 |
| 650 | 0.004 | 0.02 |

We used the MATLAB PID Tuner app in the Control System Toolbox [28] to optimize the PID controller parameters. This tool necessitates the Laplace Transform model of the process as input. Equation (11) provides a generic formula for the Laplace Transform, while Equation (12) marks the initial formula for the Laplace Transform of our specific process:

$$L[f](s) = \int_0^{\infty} e^{-sx} f(x) dx \tag{11}$$

$$L[f](s) = \int_0^{T} e^{-sx} (mx + q) dx \tag{12}$$

By leveraging the properties of integrals and Laplace Transforms, we can derive the final result:

$$L[f](s) = \left(1 - e^{-Ts}\right) \frac{m + qs}{s^2} \tag{13}$$

We employed the MATLAB PID Tuner tool to derive the PID parameters following the computational process, as presented in Tables 2 and 3.

**Table 2.** Optimal PID parameters.

| Wavelength (nm) | Setpoint (lux) | Kp | Ki | Kd |
|---|---|---|---|---|
| 450 | 4740 | 0.01 | $1.036 \times 10^{-5}$ | 1.03 |
| 650 | 1031 | 0.01 | $1.036 \times 10^{-5}$ | 1.03 |

**Table 3.** Node-RED configuration for PID controllers.

| Wavelength (nm) | Setpoint (lux) | Proportional Band | Integral Time | Derivative Time |
|---|---|---|---|---|
| 450 | 4740 | 100 | 965 | 0.0097 |
| 650 | 1031 | 100 | 965 | 0.0097 |

The tuned response is depicted in Figure 11.

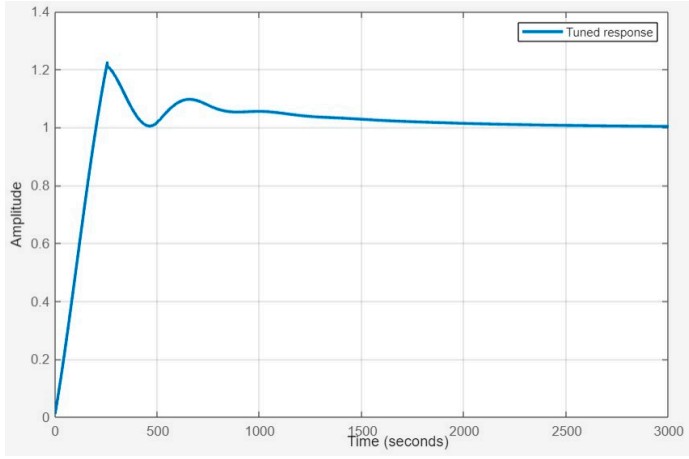

**Figure 11.** PID tuned response.

### 5.2. Optimizing Latency Performance: Migrating from Cloud to Edge

In the initial deployment phase, the Node-RED server and Mosquitto broker were hosted on a dedicated virtual machine within the Microsoft Azure public cloud computing platform. This configuration provided a baseline for performance assessment. However, recognizing the advantages of edge computing, we strategically transitioned these components to the edge, leveraging the processing capabilities of a Raspberry Pi 3.

Our evaluation primarily centered on assessing and quantifying several delay factors introduced by diverse field devices, including temperature sensors, humidity sensors, spectrometers, water level sensors, and lamps. Additionally, we conducted measurements to gauge the network round-trip time (RTT) at the application layer, capturing the total time taken for a message to travel from a sensor to the Node-RED server and back. Moreover, we examined the processing latency induced by two distinct processing environments: public cloud computing and edge computing. This analysis aimed to shed light on the individual contributions of these processing methods to overall system latency. The network RTT represents the total delay experienced by a message as it traverses the network from the sensor to the Node-RED server and back to the sensor. This delay encompasses various stages of communication, including transmission, propagation, and reception. At the outset, when a message is dispatched from the sensor, the RTT begins accumulating. As the message travels through the network, it encounters delays associated with data transmission and propagation, influenced by factors such as network congestion and distance. Upon reaching the Node-RED server, the message is processed, and the server responds by sending the processed data back to the sensor. Importantly, the RTT does not include any processing time during this round trip. Processing latency refers to the duration of time between when a message is received by the Node-RED server and when it is subsequently sent by the server. This delay encompasses the time required for the server to process the incoming message, which may involve various computational tasks, such as data analysis, decision-making, or formatting for transmission. The sensors' delay encompasses the time taken by sensors to fulfill their fundamental tasks within the system. This delay originates from the processes involved in capturing data related to environmental parameters, formatting these data into a suitable message format, and transmitting them to the designated destination.

The results in Figure 12 demonstrate a substantial 84.48% decrease in transmission latency introduced by the network after the migration to the edge—the network RTT. This shift not only enhanced the responsiveness of our system but also underscores the efficiency gains achievable by leveraging edge computing resources.

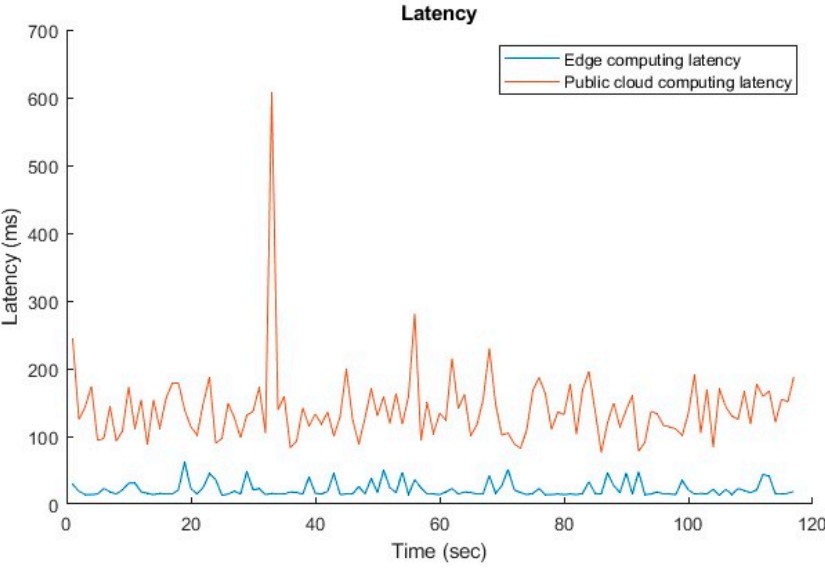

**Figure 12.** Comparative network RTT analysis.

We also measured latency introduced by processes at the edge, initially on cloud computing and subsequently on the Raspberry Pi 3 Model B at the edge. Table 4 provides a summary of the hardware components, comparing the specifications of the virtual machine installed on cloud computing with those of the Raspberry Pi 3 at the edge.

Figures 13 and 14 present an analysis of the mean processing latency in two key scenarios. Figure 13 illustrates the processing latency on the public cloud for single-message processing (e.g., water level, PID controller, temperature, and humidity). Meanwhile, Figure 14 depicts the mean processing latency on the Raspberry Pi 3 at the network's edge. Notably, transitioning computing from the cloud to the edge resulted in an increase in latency, growing from 3.34 ms to 8.11 ms.

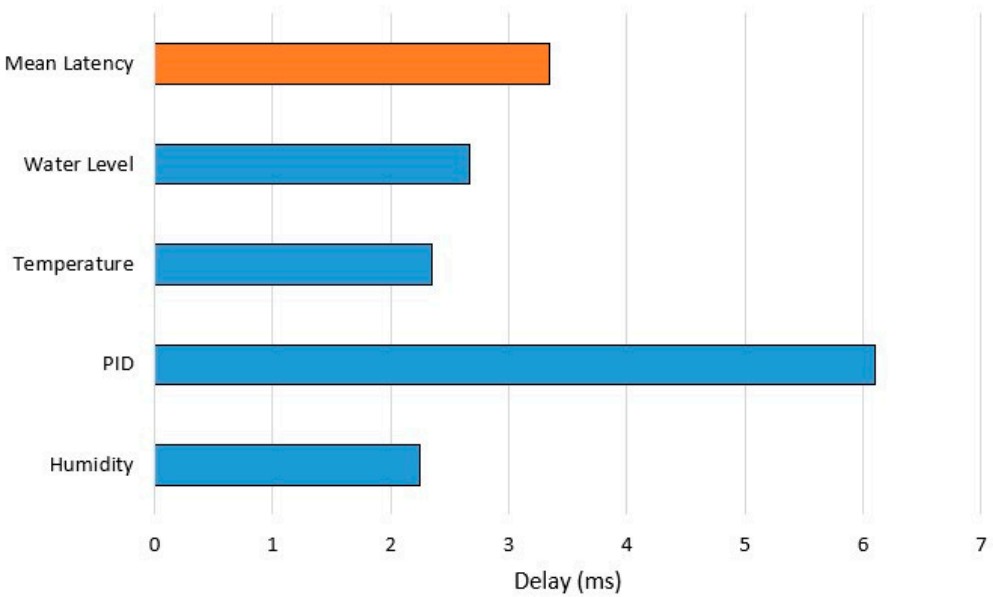

**Figure 13.** Processing latency on the public cloud.

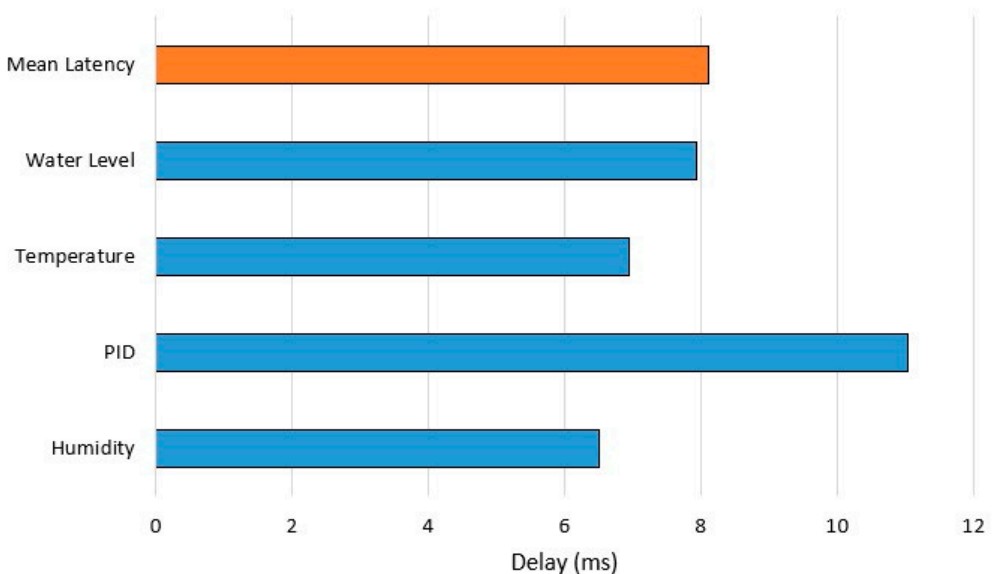

**Figure 14.** Processing latency on edge computing.

**Table 4.** Computation of the hardware processes.

| Component | No. of Cores | CPU | RAM (MB) | Operative System |
|---|---|---|---|---|
| Virtual Machine on Public Cloud | 4 | 12th Gen Intel(R) Core(TM) i7-1255U 1.70 GHz (Intel, Kiryat Gat, Israel) | 4096 | Ubuntu 20.04.6 |
| Raspberry Pi 3 Model B at the edge | 4 | Quad Core 1.2 GHz Broadcom BCM2837 (Broadcom, Palo Alto, CA, USA) | 1024 | Ubuntu 20.04.6 |

Additionally, we explored another factor contributing to service delay: measurements on smart devices equipped with sensing and actuating capabilities. Remarkably, this delay remained consistent whether data processing occurred in the public cloud or at the edge. In this case, we also measured delays for the sensing light spectrum, humidity, temperature, water level, and for actuating lamps on plants in Figure 15.

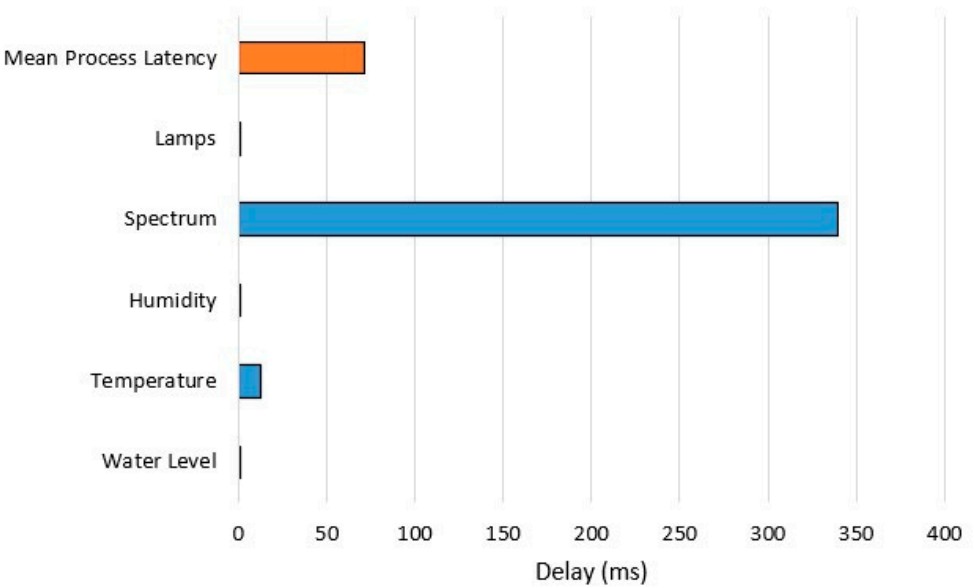

**Figure 15.** Sensing and actuating delays.

In summary, our investigation underscores that the total service latency is shaped not only by network latency but also by processing delays. The mean total latency, excluding latency from smart devices (which remains consistent whether data processing is in the public cloud or at the edge), is the sum of processing latency and network RTT means. Table 5 visually represents a remarkable 79.22% reduction in total latency when processing was shifted to the edge compared to its location in the public cloud.

**Table 5.** Latency summary.

| System Architecture | Processing Latency (ms) | Network RTT (ms) | Total Latency (ms) |
|---|---|---|---|
| Processing on Public Cloud | 3.35 | 140.70 | 144.05 |
| Processing on Edge | 8.11 | 21.83 | 29.94 |

*5.3. Monitored Parameters' Trends*

The plants were transferred from a germinator to the hydroponic greenhouse after 35 days. Throughout the trial period, the air temperature and humidity were monitored and maintained stable between 19 and 28 degrees and between 60% and 80%, respectively.

These optimal values are crucial for maximizing the yield and quality of tomato plants, as we aimed to emulate the growth period of one year [5]. Figure 16 illustrates the trends in air temperature and humidity.

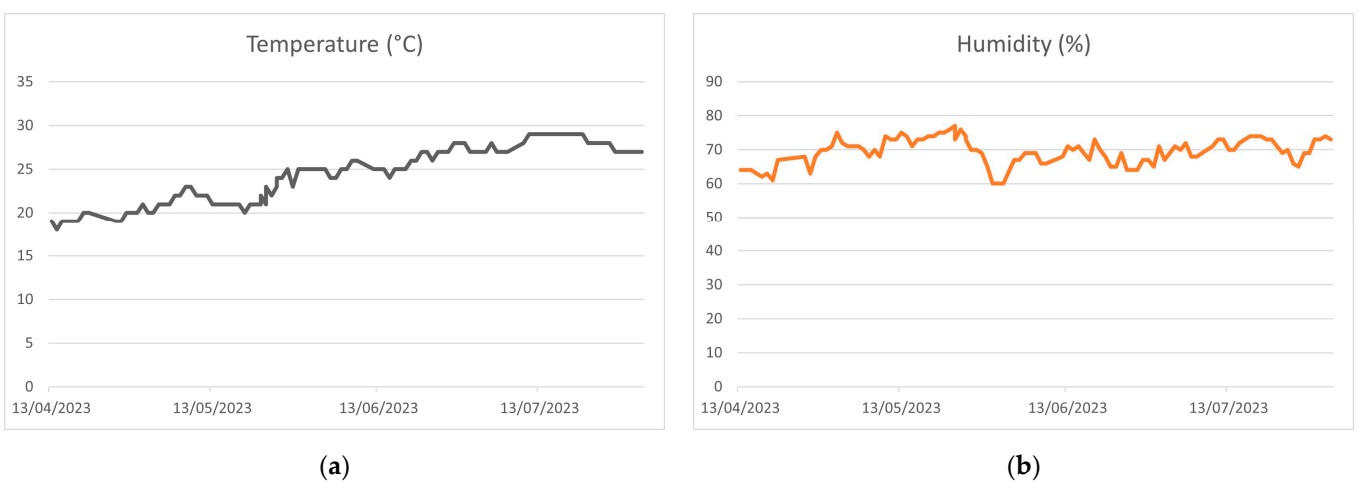

(**a**)     (**b**)

**Figure 16.** Two key trends: (**a**) the temperature trend and (**b**) the humidity trend, providing valuable insights into the environmental conditions during the experimentation period.

Monitoring the water level is essential for calculating the water uptake of the plants throughout their growth. Figure 17a presents the water uptake during the observation period (13 April to 1 August 2023) for a tomato plant.

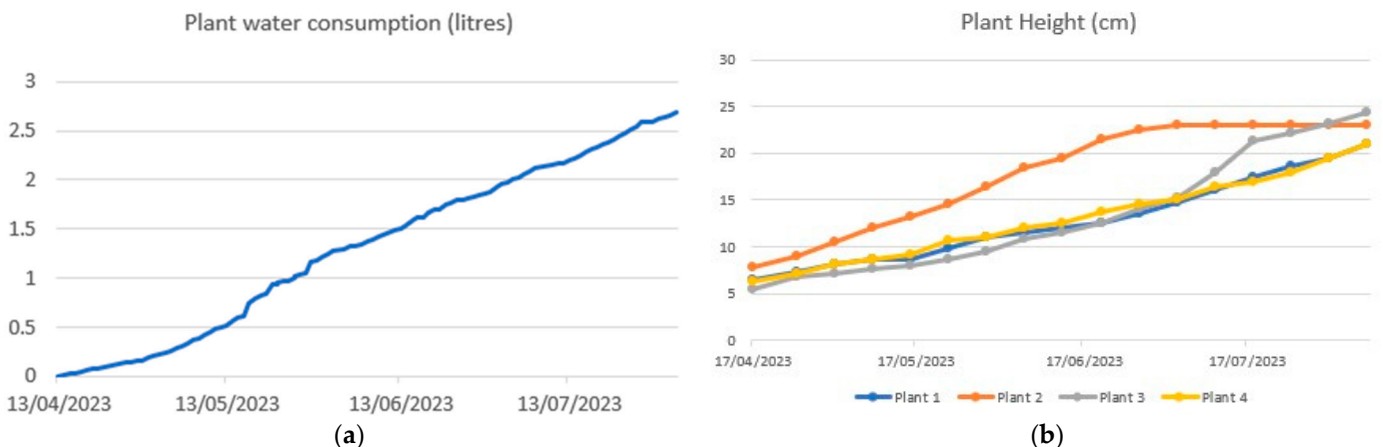

(**a**)     (**b**)

**Figure 17.** Two critical aspects: (**a**) the water consumption of a tomato plant and (**b**) the trends in plant height, providing valuable insights into the growth and resource utilization of the plants.

The light intensity is a fundamental parameter regulated by the closed-loop PID control implemented in Node-RED, ensuring an optimal environment for tomato plants. Figure 17b illustrates the growth trends of each plant, while Figure 18 showcases the fruits produced during the observation period.

At the end, the implementation of UV LED technology eliminates the presence of insects, eliminating the need for pesticides. Furthermore, the hydroponic greenhouse can incorporate eco-friendly energy sources, such as solar panels or wind blades. To optimize energy consumption in smart devices, battery-less technologies can be employed. The smart objects within the system can harness energy from the greenhouse lighting, contributing to energy efficiency. Additionally, water consumption can be minimized by replenishing the hydroponic greenhouse with rainwater stored in tanks.

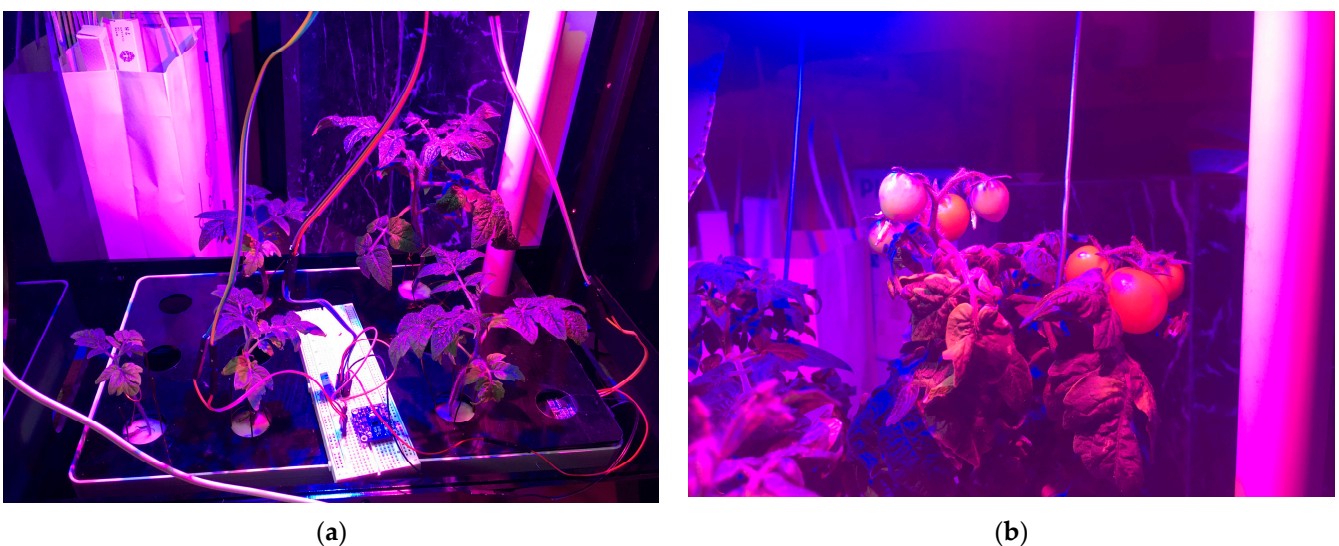

(**a**)                                                                                       (**b**)

**Figure 18.** (**a**) Tomato plant and (**b**) fruit production.

## 6. Conclusions

In this study, we presented a novel approach to hydroponic greenhouse systems by integrating Internet of Things (IoT) technologies, edge computing, and a modular architecture, as new devices or functionalities can be added without reconfiguring the existing IoT devices. Our system, GymHydro, leverages Node-RED, a low-code/no-code platform, for seamless integration of new services and devices, ensuring flexible and easy development. The implementation of a PID-controlled smart lighting system with high-power LEDs and a spectrometer enabled precise control over the light intensity in the greenhouse. The use of New Generation Service Interfaces for Linked Data (NGSI-LD) in conjunction with the MQTT protocol provided a lightweight, interoperable, and efficient communication framework for our hydroponic greenhouse. This interoperability facilitates data exchange and interaction among various components, enhancing the overall compatibility of the proposed architecture.

Our experiments involved optimizing PID controller parameters using a mathematical model of the light intensity variation concerning PWM signals provided to the LEDs. The transition of the computing infrastructure from the public cloud to the edge demonstrated a substantial reduction in transmission latency by 84.48% and in total latency by 79.22%, highlighting the benefits of edge computing in enhancing system responsiveness. Our research also aimed to understand the impact of the delay of a single device installed in a greenhouse. Here, the delay was introduced by processing with respect to the arrived message. We noticed that the delay increased in measuring the air temperature and the intensity of visible wavelengths, with respect to monitoring the air humidity and water level and to control lamps. Additionally, the processing of different arrived messages changed based on the computational components and algorithms used to process and represent data.

The GymHydro system was validated through a comprehensive trial involving the cultivation of tomato plants. Stable environmental conditions, monitored through the trial period, maintained optimal temperature and humidity levels for plant growth. The closed-loop PID-controlled smart lighting system positively influenced plant development, resulting in a healthy plant height and robust fruit production, particularly on days when sunlight was low or absent.

The integration of UV LEDs successfully eliminated the presence of harmful insects, reducing the need for chemical pesticides. Moreover, the study suggests potential strategies for eco-friendly energy sources, energy reuse, and water conservation within the hydro-

ponic greenhouse, using eco-friendly energy sources, energy-harvesting technologies (our future work), and replenishing water with filtered rainwater.

In conclusion, GymHydro represents an innovative and sustainable solution for precision agriculture, offering a scalable, modular, and technological platform for hydroponic cultivation, as well as reducing human intervention. The combination of IoT, edge computing, and PID-controlled smart lighting showcases the potential to revolutionize modern agricultural practices, promoting resource efficiency, environmental sustainability, and improved crop yields. We hope this study also serves as a practical demonstrator, providing farmers with a key tool during this crucial digital transition.

**Author Contributions:** Conceptualization, S.G., D.A. and C.B.; methodology, S.G., D.A. and C.B.; investigation, C.B.; writing—original draft preparation, C.B.; review and editing, S.G. and D.A.; supervision, D.A.; project administration, S.G. All authors have read and agreed to the published version of the manuscript.

**Funding:** This research received no external funding.

**Data Availability Statement:** Data is contained within the article.

**Acknowledgments:** This work was partially supported by the Italian Ministry of Research (MUR), in the framework of the CrossLab and ForeLab Projects (Departments of Excellence), and by the National Center for the Technologies in Agriculture, AGRITECH.

**Conflicts of Interest:** The authors declare no conflicts of interest.

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
