# Peer review of "GymHydro: An Innovative Modular Small-Scale Smart Agriculture System for Hydroponic Greenhouses"

_electronics, doi:10.3390/electronics13071366_

Round 1

Reviewer 1 Report

Comments and Suggestions for Authors

I find a paper entitled "GymHydro: An Innovative Modular Small-Scale Smart Agriculture System for Hydroponic Greenhouses" to be of particular interest, well written and having a potential for contributing to the science.

I have only 2 observations:

1) Literature review could be enhanced. This would help to better entrench the current research in the scientific vacuum this paper aims to fulfill.

2) Results should be better juxtaposed with the current theoretical streams

Author Response

Thank you for your thoughtful review and positive feedback on our manuscript.

1 - We have taken note of your observation regarding the literature review and have made enhancements to provide a more thorough contextualization of our research within the existing scientific literature. This revision aims to strengthen the foundation of our study and better align it with the current research landscape in the field. We are grateful for your advice.

2 - Regarding your second observation about juxtaposing the results with current theoretical streams, we appreciate your suggestion. However, upon careful consideration, we believe that our focus on presenting our results follows the “Methodology” section, which includes the description of choices of components, plant absorption wavelengths, PID algorithm, and conclusion with a description of the dashboard. Additionally, the “System Setup and Functional Validation” section begins with the circuit diagram of adopted components, optimization of PID controllers, analysis of various architecture and device delays, and concludes with key information that can be visualized on our dashboard. (components, controllers, delays and dashboard)

We are grateful for your valuable feedback, which has contributed to the improvement of our manuscript.

Thank you once again for your time and consideration.

Reviewer 2 Report

Comments and Suggestions for Authors

The idea of the paper "GymHydro: An Innovative Modular Small-Scale Smart Agriculture System for Hydroponic Greenhouses" is interesting because it tries to solve the problems faced by modern agriculture, such as the rational use of resources in agricultural production (water, pesticides, etc.) in extreme weather conditions (especially prolonged droughts).

The novelty of the proposed solution in relation to classic cultivation in hydroponic systems is the introduction of a modular platform for regulating and controlling plant cultivation conditions (air temperature, air humidity, water levels under plant roots, pH levels, and light intensity). This modular platform enables the automation of parameter regulation, which facilitates work and increases the efficiency of growing agricultural products. In addition, the platform enables the addition of new devices and services without redesigning and changing the existing system. Another novelty is using the LoRa interface in the 2.4 GHz ISM band, which enables low latency applications, worldwide roaming, and global interoperability.

The references used in the paper are precise and appropriate.

The research conclusions are consistent with the evidence and arguments presented through the discussion in the paper. In conclusion, the authors stated that GymHydro represents an innovative and sustainable solution for precision agriculture, offering a scalable, modular, and technological platform for hydroponic cultivation. Using the Node-RED platform (a low-code/no-code platform) enables simple integration of new devices and services into the existing system, demonstrated through the transition from one MCU node to multiple nodes, enhancing its capability to address real-world use cases. Using New Generation Service Interfaces for Linked Data (NGSI-LD) with MQTT protocol provides a lightweight, interoperable, and efficient communication framework for our hydroponic greenhouse. The transition of the computing infrastructure from the public cloud to the edge demonstrated a substantial reduction in transmission latency and total latency, highlighting the benefits of edge computing in enhancing system responsiveness.

I have two objections to the presentation of the research results:

1) The research shows no comparative display of cultivation within the GymHydro platform and the classical hydroponic systems. It would be good if the authors, in addition to growing tomatoes within the GymHydro platform, did a test of cultivation in a classic hydroponic system, after which they could compare the quality and quantity of tomato fruits grown in both systems and in this way further demonstrate the advantages of the GymHydro platform. It would be desirable to do this for several different plant cultures.

2) It would be desirable to increase the quality of the images where possible. For example, Figure 1 is of poor quality, and the details are unclear.

Pay attention to the numbers of the figure caption because the numbers 12 and 13 are repeated.

There is no reference to Figure 11 in the text.

Correct the numbers in the figure captions and the references to figures in the text.

Author Response

Thank you for your insightful review of our paper. We are glad to hear that you find the idea of our paper interesting. We appreciate your positive comments regarding the novelty of our proposed solution, particularly the introduction of a modular platform for regulating and controlling plant cultivation conditions. The automation capabilities of this platform and its flexibility for adding new devices without redesigning the system are key features we aimed to highlight in our work.

1 - Regarding your observations, we understand your concern about the lack of comparative display of cultivation within the GymHydro platform and classical hydroponic systems. We agree that conducting tests across different plant cultures in both systems would provide valuable insights and strengthen our demonstration of the advantages of GymHydro. We have planned this work in the near future, but as you are surely aware, we need longer timelines to conduct it and achieve results, so we cannot include it in this manuscript.

2- We also take note of your comment regarding the quality of images and the numbering issues in figure captions and references. We deeply apologize for the oversight, and we have immediately corrected everything.

Once again, we appreciate your thorough review and constructive feedback.

Thank you for your time and consideration.

Reviewer 3 Report

Comments and Suggestions for Authors

1-      On line 64, within the estimated estimates, a measurement must be placed to measure the level of nutrients in the medium

2-      In line 85, please list the most important results obtained through their study

3-      In line 89, I would like to list the most important results obtained through their study

4-      In line 101, please put the name of the application for Android as an example application for experimentation

5-      Conclusion: More focus is needed on outputs

6-      References are few and more are needed

7-      In line 164, I hope to explain more than what scholars 10, 11, and 12 have done

8-      In line 166, please clarify the range from to as was done in the study of scientist 14

9-      In line 253, please explain the time range used compared with each wavelength that was estimated

10-   Have confirmatory experiments been conducted on other plants, even on fast-growing leafy plants?

11-   In line 343, is it possible to put an indicator of the growth rate of the plant in accordance with time and environmental components?

Author Response

Thank you for taking the time to review our manuscript. We appreciate your valuable feedback and suggestions for improving our work.

Below are our responses to each of your comments:

1,2,3,4,6 – We have taken note of your observation regarding the literature review and have made enhancements to provide a more thorough contextualization of our research within the existing scientific literature.

5 – We have modified and enhanced the “Conclusion” section, providing a more detailed description of our results.

7, 8 – We have tried to add some details regarding the works, describing their operations more thoroughly.

9 - We have used a time range of a minute, which was compared with each estimated wavelength to analyze their respective impacts.

10 – Unfortunately, confirmatory experiments have not been conducted on other plants, including fast-growing leafy plants. However, it is an excellent idea to consider for future experimentation to compare the growth of other plants, even with shorter experiment durations. Thank you for the suggestion; we were not aware that such plants existed.

11 - Certainly, it is possible to add any type of indicator to the dashboard easily and quickly by adding a few blocks to the Node-RED flow on the server. One of the objectives of our work is to be able to add or modify services quickly to adapt them to individual users thanks to the no-code/low-code paradigm. However, in this case, I don't think it's useful because plant height was measured manually, so we don't have an automatic system that can provide the data to be processed and displayed on the dashboard. Nonetheless, developing an automatic plant height monitoring system is part of our future work, especially utilizing neural networks. It will also be possible to integrate it seamlessly into the existing systems on the prototype because it was designed for this purpose, gradually adding systems to increase automation using a single interface without modifying configurations to already functioning services or devices.

Your feedback is highly valuable to us, and we are grateful for the opportunity to enhance our work based on your insights.

Thank you once again for your thorough review and constructive comments.

Reviewer 4 Report

Comments and Suggestions for Authors

1. [Line 13] In the Introduction section, there is a lack of clarity on the specific research gap or problem that this study aims to address. Please provide a clear statement of the research problem and the motivation behind developing the proposed system before the sentence 'This paper introduces a novel system with a modular architecture ...'

2. [Section 2. Related Works] The Related Works section is rather brief and does not provide a comprehensive overview of the existing literature in the field. You're supposed to follow the scientific conduction process how you identify this issue from previous literatures. The current content contains insufficient work. It would be beneficial to include a more in-depth discussion of relevant studies, highlighting their strengths, limitations, and how your work differs or contributes to the existing body of knowledge.

3. [System Architecture, Lines 124-127] In the System Architecture section, the rationale behind the choice of specific technologies and protocols, such as LoRa, MQTT, and NGSI-LD, is not clearly explained.

Neither the modular approach is not clear. You mentionedmodular approach is presented as a key feature of the proposed system. However, your paper lacks a detailed explanation of above-mentioned objects. You need to provide a more detailed justification for these choices in an appropriate aspect (not all aspects).

4. [Section 4.1] In the Smart Lighting with PID Controller subsection, the rationale behind the choice of specific LED wavelengths and their relevance to tomato plant growth is not clearly justified. Please provide more detailed information on the scientific basis for these choices and their potential impact on plant growth and yield. Also, in lines 295-326, the description of the PID controller implementation lacks clarity. Please elaborate more.

5. [4.2 Dashboard, Lines 326-339] Note this comment is a suggestion, not compulsory:

The Dashboard subsection is a little bit brief and lacks information on the user interface design considerations, the rationale behind the chosen visualization techniques, and the potential benefits for end-users. If it's not necessary (provided your work doesn't consider UI in a high weight), you also need to explain the reason they are not in the high priority to be discussed.

6. [Figures 12-14] The figures in the Monitored Parameters Trends subsection are not clearly labeled or captioned, making it difficult to interpret the data presented. Please ensure that all figures are properly labeled and include descriptive captions.

Comments on the Quality of English Language

Minor editing of English language required

Author Response

Thank you for your detailed feedback on our manuscript. We appreciate the time and effort you've taken to review our work and provide valuable insights.

Below are our responses to each of your points:

1 - We acknowledge the need for a clear statement of the research problem and the motivation behind developing our proposed system in the “Abstract” section. We have revised the section to provide a explicit description of the research gap and the problem our study aims to address before introducing the novel system.

2- Thank you for highlighting the need for a more comprehensive overview of existing literature in the “Related Works” section. We have made enhancements to provide a more thorough contextualization of our research within the existing scientific literature.

3 - We understand the importance of explaining the rationale behind our choice of specific technologies and protocols. In the “System Architecture” section, we have provided a description of the components utilized in our architecture. However, a more detailed justification for the selection of LoRa, MQTT, NGSI-LD, and the modular approach is provided in the 'Methodology' section. This detailed justification is presented when we describe the singular choices of our components and protocols adopted in comparison to other possible solutions.

4- Your feedback regarding the rationale behind the choice of LED wavelengths and the clarity of the PID controller implementation in the Smart Lighting with PID Controller subsection is valuable. We will add detailed information on the scientific basis for our choices.

5- Thank you for your suggestion regarding the Dashboard subsection. Our aim was to demonstrate how the approach used in our work makes it easy to implement simple and intuitive dashboards, with the ability to modify them easily based on individual user needs. The focus was primarily on showcasing the technical aspects of our system rather than delving deeply into user interface design considerations and visualization techniques. However, we understand the importance of these aspects.

6- Thank you for addressing the labeling and captioning issues in the figures of the Monitored Parameters Trends subsection. We will make sure that all figures are appropriately labeled and include descriptive captions to enhance data interpretation. We sincerely apologize for this oversight.

Thank you once again for your thoughtful evaluation and constructive suggestions.

Round 2

Reviewer 4 Report

Comments and Suggestions for Authors

The authors have adequately addressed all identified comments from the last round review.

The work conveys novel contributions advancing the topic in their research field which will provide value to readers that merits journal dissemination.

Thanks for inviting me as the reviewer.